# 3D Echo Characterization of Proportionate and Disproportionate Functional Mitral Regurgitation before and after Percutaneous Mitral Valve Repair

**DOI:** 10.3390/jcm11030645

**Published:** 2022-01-27

**Authors:** Sara Cimino, Luciano Agati, Domenico Filomena, Viviana Maestrini, Sara Monosilio, Lucia Ilaria Birtolo, Michele Mocci, Massimo Mancone, Gennaro Sardella, Paul Grayburn, Francesco Fedele

**Affiliations:** 1Department of Clinical, Internal, Anesthesiological and Cardiovascular Sciences, Sapienza University of Rome, Policlinico Umberto I, 00161 Rome, Italy; sara.cimino@uniroma1.it (S.C.); luciano.agati@uniroma1.it (L.A.); domenico.filomena@uniroma1.it (D.F.); viviana.maestrini@uniroma1.it (V.M.); sara.monosilio@gmail.com (S.M.); ilariabirtolo@gmail.com (L.I.B.); michelemocci@hotmail.com (M.M.); rino.sardella@uniroma1.it (G.S.); francesco.fedele@uniroma1.it (F.F.); 2Division of Cardiology, Department of Internal Medicine, Baylor University Medical Center, Dallas, TX 75246, USA; paul.grayburn@BSWHealth.com

**Keywords:** PMVr, MitraClip, EROA/LVEDV ratio, functional mitral regurgitation, disproportionate MR

## Abstract

Background: The impact of percutaneous mitral valve repair (PMVr) on long-term prognosis in patients with functional mitral regurgitation (FMR) is still unclear. Recently, a new conceptual framework classifying FMR as proportionate (P-MR) and disproportionate (D-MR) was proposed, according to the effective regurgitant orifice area/left ventricular end-diastolic volume (EROA/LVEDV) ratio. The aim was to assess its possible influence on PMVr efficacy. Methods: A total of 56 patients were enrolled. MV annulus, LV volumes and function were assessed. Global longitudinal strain (GLS) was also calculated. Patients were divided into two groups, according to the EROA/LVEDV ratio. Echocardiographic follow-up was performed after 6 months, and adverse events were collected after 12 months. Results: D-MR patients (n = 28, 50%) had a significantly more elliptical MV annulus (*p* = 0.048), lower tenting volume (*p* = 0.01), higher LV ejection fraction (LVEF: 32 ± 7 vs. 26 ± 5%, *p* = 0.003), lower LVEDV, LV end-systolic volume (LVESV) and mass (LVEDV/i: 80 ± 20 vs. 126 ± 27 mL, *p* = 0.001; LVESV/i: 60 ± 20 vs. 94 ± 23 mL, *p* < 0.001; LV mass: 249 ± 63 vs. 301 ± 69 gr, *p* = 0.035). GLS was more impaired in P-MR (*p* = 0.048). After 6 months, P-MR patients showed a higher rate of MR recurrence. After 12 months, the rate of CV death and rehospitalization due to HF was significantly higher in P-MR patients (46% vs. 7%, *p* < 0.001). P-MR status was strongly associated with CV death/rehospitalization (HR = 3.4, CI 95% = 1.3–8.6, *p* = 0.009). Conclusions: Patients with P-MR seem to have worse outcomes after PVMr than D-MR patients. Our study confirms the importance of the EROA/LVEDV ratio in defining different subsets of FMR based on the anatomical characteristic of MV and LV.

## 1. Introduction

Functional mitral regurgitation (FMR) occurs in patients with left ventricular (LV) dysfunction in the absence of a significant structural abnormality of mitral valve leaflets [1]. Understanding the causal and temporal relation between MR and the grade of LV dysfunction can be challenging, because both pathologies are associated with volume overload, progressive LV dilation and heart failure (HF) progression with symptoms worsening and poor clinical outcome [2,3,4].

Since FMR is mainly attributable to LV dysfunction, MV surgical treatment alone has not been shown to improve death and hospitalization for HF [5].

More recently, percutaneous mitral valve repair (PMVr) with the MitraClip system has become an option for patients with FMR with high surgical risk, who remain symptomatic despite appropriate optimal medical and/or device therapy [6]. The effectiveness of PMVr was demonstrated in patients with degenerative MR [7], but the long-term prognostic impact in FMR patients is still debated. In spite of that, current criteria for FMR candidates to MitraClip selection are still based on MV characteristics [6] irrespective of LV, which is paradoxically the target to treat [4]. Notably, the recently published Cardiovascular Outcome Assessment of the MitraClip percutaneous Therapy for Heart Failure (COAPT) and Mitra-FR trials showed apparently opposite results [8,9] in terms of cardiovascular (CV) death and rehospitalization for HF, which are favorable in COAPT trial but not in Mitra-FR. Critical analysis of the two randomized trials [10] revealed substantial differences in the populations enrolled in Mitra-FR and COAPT. Specifically, Mitra-FR patients presented with less severe MR and more dilated LV if compared to COAPT patients. Subsequently, Grayburn et al. [11] proposed a conceptual framework that explains the different outcomes in COAPT and Mitra-FR, depending on whether the effective regurgitant orifice area (EROA) was proportionate (P-MR) to the left ventricular end-diastolic volume (LVEDV) or not (disproportionate or D-MR). In P-MR, the mechanism of FMR can be explained by LV dilation alone, while in D-MR the degree of MR is more often due to focal regional wall motion abnormalities or dyssynchrony, factors that are not likely to respond only to medical therapy. Although this hypothesis helps to explain differences between Mitra-FR and COAPT trials, it has not been evaluated prospectively in patients.

Recently, technological advancement in three-dimensional (3D) echocardiography techniques allows for more precise and reliable evaluation of both MV and LV features, mainly due to the employment of semi-automated or fully automated quantification [12].

Accordingly, the aim of the present study was: (1) to verify if this new proposed model of MR classification could be applied in clinical practice and could reveal any difference in outcomes after MitraClip implantation between patients with P-MR and D-MR; (2) to describe 3D echocardiographic characteristics of P-MR and D-MR patients; (3) to identify possible echocardiographic predictors of MitraClip success.

## 2. Methods

Study population: Fifty-six consecutive patients referred to our department for percutaneous mitral valve repair with MitraClip implantation were enrolled prospectively in this study. Inclusion criteria were: (1) severe FMR, (2) reduced LV ejection fraction (LVEF <45%); (3) high surgical risk, excluding traditional surgery; (4) symptoms of HF despite optimal medical/device therapy where indicated and revascularization (5) NYHA class III or IV and (6) high-quality echocardiographic windows. Exclusion criteria were: (1) degenerative MR, (2) MV morphological properties that would make MitraClip implantation unlikely or unsuitable [3,13,14] low life expectancy. The MitraClip procedure was explained to the patients, as well as alternative options (medical treatment or high-risk MV surgery). The “Heart Team” evaluated patients, and conventional surgery was excluded because of excessive morbidity and mortality (high Logistic Euro Score or STS score, or excessive comorbidities not in traditional scores) [13]. The local Ethics Committee approved the present study, and all patients provided written informed consent.

The baseline and follow-up functional status was assessed according to the New York Heart Association criteria. TTE follow-up was performed after 6 months. MR recurrence was defined as residual MR grade moderate-to-severe or severe as assessed through 6-month FU echocardiogram. A clinical follow-up was performed after 12 months to record the occurrence of cardiovascular death and rehospitalization for HF.

Echocardiography: All patients enrolled underwent transthoracic (TTE) and transesophageal (TOE) two- and three-dimensional echocardiography (Philips X5-1 and Philips 7-xt Transducers, EPIQ7C) before and after the procedure of percutaneous mitral valve repair. TOE post-operative evaluation was performed in the majority of cases in the cath-lab, at the end of the procedure. The presence of MR at baseline was qualified by color Doppler and quantified by the vena contracta width and the proximal isovelocity surface area method in accordance with the current practice guidelines [15]. All patients were assigned an MR severity score in four grades, from 1 (mild) to 4 (severe), according to the quantitative measure of the EROA and regurgitant volume, as assessed by TOE. The regurgitant volume was estimated as the EROA multiplied by the velocity time integral of the regurgitant jet. Procedural success was defined as the reduction of the MR severity score to 2 or less after clip implantation. To note, since the study was conducted before the year 2021, the judgment on MR severity was based on the previous ESC guidelines [6].

The following parameters were considered in baseline and post-operative evaluation: LV end-diastolic and end-systolic volume indexed to body surface area (LVEDV/i and LVESV/i, respectively), LV ejection fraction (LVEF), obtained using three-dimensional echocardiography (3DE) (Heart Model by Philips, fully automated LV evaluation), LV mass. Right ventricular dimension, function, and pulmonary artery systolic pressure (PASp) were also assessed by 2D TTE. Two-dimensional speckle tracking analysis with global longitudinal strain (GLS) was also obtained in all patients (Philips, QLAB, v. 13, AutoStrain supplied by Tomtec). Mitral valve annulus assessment was performed by 3D TOE before and immediately after the procedure (Philips, QLAB v. 13, mitral valve quantification (MVQ)). The following parameters were obtained with MVQ analysis: [1] annulus ellipticity, [2] annulus antero-posterior (AP) diameter; [3] tenting volume; [4] tenting height; [5] anterior leaflet (AL) and posterior leaflet (PL) angle, as previously described [16].

After a complete echocardiographic evaluation, including both MR and LV quantification, patients were divided into two groups according to the definition of P-MR and D-MR based on the graph illustrating the EROA/LVEDV (mm^2^/mL) ratio, as previously described [11]. We used the previously published cut-off value of 0.14 obtained through extrapolation from the graph after critical analysis of COAPT patients: patients with an EROA/LVEDV ratio >0.14 were classified as D-MR, and patients with a ratio ≤ 0.14 were classified as P-MR [17].

## 3. Reproducibility

Intra-observer and inter-observer variability for the 3D automatic measurements of LVEF and LV volumes was assessed in a sample of 10 patients, as well as 3D manual measurements of annular dimensions and EROA estimation. Two investigators measured blinded the same echo loops, and one investigator repeated the analysis one week later, blinded to the previous measurements, as previously published by our research laboratory [18,19].

## 4. MitraClip Procedure

All patients underwent endovascular edge-to-edge MV repair, as previously described [13,14,18]. All procedures were performed using the 24-French MitraClip device (Abbott Vascular, Santa Clara, CA, USA) by the same interventional cardiologist. All clips were implanted under general anesthesia and the procedures were TOE guided. Hemostasis was achieved by compression of the vein for 12 h. Patients were treated with double antiplatelet therapy after the intervention and oral anticoagulants where indicated.

## 5. Statistical Analysis

Continuous variables were presented as mean ± standard deviation (SD) if normally distributed, or median and extremes if not. Normality distribution was assessed using the Kolmogorov–Smirnov test. Differences between groups were assessed using Student’s t test or the Mann–Whitney rank sum test for unpaired comparisons, as appropriate.

The mean differences between the baseline parameters in the two groups are reported. The categorical variables are expressed as counts and percentages and were compared using the Chi-square test or Fisher exact test, as appropriate. Differences were considered statistically significant when *p* < 0.05. Univariable Cox proportional-hazards regression analysis was used to identify factors associated with events. Variables were used in the model as continuous or categorical variables when possible. The Kaplan–Meier method was used for cumulative survival analysis with the log-rank test for assessing statistical differences between curves. A value of *p* < 0.05 was considered to indicate statistical significance. Statistical analyses were performed using the Statistical Package for Social Sciences, version 23.0 (SPSS, Chicago, IL, USA). Inter-class correlation coefficients (ICC) were calculated to assess inter-observer and intra-observer agreement of 3D echocardiography measurements.

## 6. Results

Descriptive analysis: Mean age of the whole population was 73 ± 7 years, and 73% were male. Mean Logistic EuroScore was 15.5 ± 10, and mean STS score was 7.2 ± 8. Twenty-eight (50%) patients had an ischemic etiology of LV dysfunction and thus FMR. All patients were in NYHA class III-IV despite being on optimal medical therapy, including device treatment where indicated. All baseline characteristics are shown in Table 1. Applying the cut-off value EROA/LVEDV >0.14, P-MR was detected in 28 patients (50%) and D-MR in the remaining 28. All differences between groups are shown in Table 1. At baseline, D-MR presented with a significantly more elliptical MV annulus (ellipticity 163 ± 9% vs. 139 ± 33, *p* = 0.048), with lower tenting volume (3.2 ± 1.2 vs. 6.3 ± 2.3 mL, *p* = 0.01) and height (6.5 ± 1.4 vs. 9.6 ± 2.4 mm, *p* = 0.01). Despite the absence of significant differences in PL and AL angle between the two groups, we observed slightly higher values of PL angle and lower values of AL angle in D-MR patients (respectively, 48 ± 12 vs. 46 ± 10°, *p* = 0.6; 24 ± 4 vs. 29 ± 8°, *p* = 0.08). Examples of MVQ analysis in D-MR and P-MR patients are shown in Figure 1A,B1,B2. TOE loops of D-MR and P-MR are shown in Appendix A.

Moreover, D-MR presented with higher LVEF (32 ± 7 vs. 26 ± 5%, *p* = 0.003), while LVEDV/i, LVESV/i and LV mass were higher in P-MR (LVEDV/i 80 ± 20 vs. 126 ± 27 mL, *p* = 0.001; LVESV/i 60 ± 20 vs. 94 ± 23 mL, *p* < 0.001; LV mass 249 ± 63 vs. 301 ± 69 gr, *p* = 0.035, respectively). Examples of 3D LV analysis in D-MR and P-MR are shown in Figure 2A,B and Appendix A. EROA was slightly higher in D-MR, although this difference was non-significant (0.39 ± 0.1 vs. 0.33 ± 0.1, *p* = 0.7). GLS was slightly higher in P-MR compared to D-MR, despite both being severely impaired (−9.5 ± 4.1% vs. −7.2 ± 2.7%, *p* = 0.048). In Appendix A, an example of GLS analysis is depicted. Figure 3 shows differences in LVEDV, LVESV, tenting height and tenting volume in D-MR and P-MR.

Outcome analysis: Six months from MitraClip implant, P-MR patients showed a higher rate of MR recurrence at 6 months (14% vs. 57%, *p* < 0.001). At 12-month follow-up, CV death and rehospitalization due to HF rate were significantly higher in P-MR (46% vs. 7%, *p* < 0.001). Kaplan–Meier curves (Figure 1) demonstrated a significantly higher event-free rate in D-MR, with log-rank *p* < 0.001 (Figure 4A) and in patients with lower LVEDV/i and lower tenting volume (inferior to their median values) *p* = 0.048 and *p* = 0.018 (Figure 4B,D). On the contrary, survival analysis for PASp values did not show any significance (log-rank *p* = 0.069) (Figure 4C). At Cox univariate analysis, P-MR status was strongly associated with CV death/HF rehospitalization with 3.4 (1.3–8.6) (*p* = 0.009), as shown in Table 2.

Reproducibility analysis: Intra-observer agreement analysis showed an ICC of 0.981 (*p* < 0.001, 95% CI = 0.92–0.996) for LVEF measurements, 0.996 (*p* < 0.001, 95% CI = 0.985–0.999) for LVEDV/i measurements, 0.998 (*p* < 0.001, 95% CI = 0.992–0.999) for LVESV/i measurements and 0.998 (*p* < 0.001, 95% CI = 0.993–0.999) for MVQ analysis. Inter-observer agreement analysis showed an ICC of 0.938 (*p* < 0.001, 95% CI = 0.75–0.996) for LVEF measurements, 0.994 (*p* < 0.001, 95% CI = 0.736–0.999) for LVEDV/i measurements, 0.997 (*p* < 0.001, 95% CI = 0.986–0.999) for LVESV/i measurements and 0.996 (95% CI = 0.998–0.999) for MVQ analysis, as shown in Table 3.

## 7. Discussion

The present study confirmed that: (1) the model of MR characterization on the basis of the EROA/LVEDV ratio [11,12,13,14,15,16] is applicable in clinical practice; (2) P-MR patients had higher LV volumes, less elliptical MV annulus and higher tenting volume as assessed by 3D echo; (3) P-MR patients had worse prognosis in terms of CV death and HF hospitalization within 1 year along with a lower rate of MV repair durability; (4) no other relevant clinical and echocardiographic features demonstrated association with outcomes over P-MR status.

This is the first prospective clinical study that applies Grayburn’s model [11,12,13,14,15,16] in a population of FMR patients using 3D echo and confirming its relationship with the success of MitraClip procedure.

The prognostic impact of MitraClip intervention is currently debated, since the results of the two prospective clinical trials, focused on enhancing optimal medical therapy with MitraClip procedure in patients with FMR, showed conflicting results. In detail, the COAPT trial demonstrated that MitraClip treatment reduces the rate of hospitalization for HF and improves 2-year survival in selected patients with FMR compared with optimal medical therapy alone [8]. Conversely, the Mitra-FR trial failed to demonstrate significant improvement in clinical outcomes after MitraClip implantation in patients with FMR at 12-month follow-up, in terms of death and hospitalization rate [9]. The role of MitraClip therapy in reducing the cardiac overload from severe MR and improving symptoms is widely recognized [20,21], but data on long-term prognosis are not always encouraging [22,23].

Even before the publication of these two trials [8,9], some data demonstrated substantial differences between FMR patients who could have more benefit after MitraClip and patients who did not. Previously we observed that patients with a more dilated LV and MV annulus were less likely to benefit from the procedure in terms of MV repair durability over time [18]. We observed later that patients with a less dilated LV and lower PASp were more likely to experience a significant reverse LV remodeling at follow-up [19]. Prognostic data were not available at the time of publication of these data, and the prognostic role of these parameters was not assessed.

To explain the different outcomes of patients between COAPT e Mitra-FR, it has been pointed out that LVEDV/i was higher in MITRA-FR (135 ± 37 mL/m^2^) compared to patients in COAPT (101 ± 34 mL/m^2^) trial. However, in both studies, these left ventricles would be considered severely dilated according to guidelines [24], which makes it less credible that the LV volume alone explains the different prognoses [25]. In addition, it was speculated that negative results from the Mitra-FR trial could be explained by the fact that patients had more advanced HF, while COAPT patients underwent the intervention at the “right time” during the course of the disease. This hypothesis does not seem plausible because in both control groups, the all-cause mortality rate was similar and, paradoxically, NT-Pro-BNP values were higher in COAPT patients [25]. Furthermore, COAPT patients presented more severe MR at baseline, because severity was assessed according to US guidelines, with higher EROAs [10,26]. On the basis of the ratio between EROA and LVEDV, Grayburn et al. [11,16] observed that COAPT patients could be approximately classified as D-MR, presenting with EROA about 30% higher and LV volumes about 30% smaller than Mitra-FR patients, which, conversely, could be classified as P-MR.

According to our results, this model is likely to be reliable and applicable in clinical practice. D-MR patients are identified by less dilated LV and MV annulus with a smaller tenting volume. Moreover, D-MR patients have less symmetric tethering of valve leaflets than P-MR.

Accordingly, the recently published Euro-SMR study [27] observed a lower 2-year mortality rate after PMVr in patients with FMR who were defined as “MR dominant” and “MR-LV co-dominant” when compared with “LV dominant”. These three groups were always obtained from the EROA/LVEDV ratio. On the contrary, Messika-Zeitoun et al. [28], in a post-hoc analysis of the Mitra-FR trial, did not observe, even after different patients’ classification based on the EROA/LVEDV ratio, a subset of patients that have benefited from PMVr using the MitraClip system. The authors hypothesized that other non-explored parameters such as the right ventricular function, LV fibrosis and LV contractile reserve possibly different between the two trials’ (Mitra-FR and COAPT) populations might also explain the observed divergent outcomes. Accordingly, our results showed that there are other parameters associated with worse outcomes such as tenting volume and PASp.

In our population, patients classified as D-MR had a lower rate of CV death and HF rehospitalization as long as repair durability compared with the P-MR group. It is important to underline that the employment of the 3D echocardiographic technique increases the reliability and reproducibility of LV volume assessment [29]. Grayburn and Packer have proposed the conceptual “proportionality” framework based on group data and have questioned whether it can be applied in individual patients because of the high variability of 2D echocardiographic measurements [16]. Three-dimensional (3D) echocardiographic measurements are more reproducible and compare more favorably to magnetic resonance imaging for measuring LV volumes and LVEF [30].

## 8. Limits

A major limitation of the study is the small sample size that could mask other differences between groups and does not allow performing a multivariable analysis. Invasive data such as invasive LA pressure data are not available. Follow-up observation is limited, and the small number of events during follow-up precludes identifying the impact of CV death and rehospitalization for HF as separated end-points. Finally, there is no medical control group for comparison. Cardiac magnetic resonance study with LV fibrosis detection could be an important adjunctive value for future studies.

## 9. Conclusions

The findings of our study support the use of MR classification in clinical practice based on the EROA/LVEDV ratio proposed by Grayburn et al. to improve the selection of patients that could benefit from MitraClip implant beyond the evaluation of valve anatomy. In our cohort of patients, P-MR represents a class of patients at higher risk of CV events at 1-year follow-up after MitraClip intervention above other echocardiographic parameters considered in isolation as LV volume, ejection fraction, strain, MR severity and geometric evaluation of the MV annulus. Further longitudinal studies in wider cohorts are necessary to confirm this data and to clarify the discordant results of COAPT and Mitra-FR trials. However, waiting for RESHAPE-HF2 trial results, these findings could be helpful to improve the selection of patient candidates to MitraClip.

## Figures and Tables

**Figure 1 jcm-11-00645-f001:**
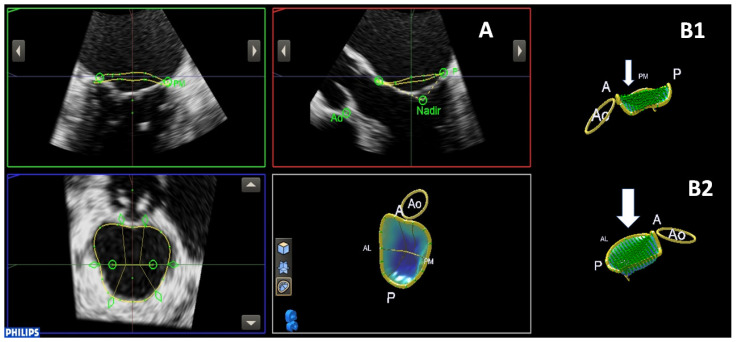
(**A**) Mitral valve quantification analysis: an example of MVQ analysis showing bi-commissural view (**upper-left box**), left ventricular outflow tract view (**upper-right box**), MV annulus in short axis (**lower-left box**) and 3D reconstruction of the MV annulus and leaflets (**lower-right box**). (**B1**) Three-dimensional (3D) reconstruction of the MV annulus and leaflets in D-MR patients, where the green area represents the tenting volume; (**B2**). Three-dimensional (3D) reconstruction of the MV annulus and leaflets in P-MR patients, where the green area represents the tenting volume, which is higher if compared with D-MR and labeled by a big white arrow.

**Figure 2 jcm-11-00645-f002:**
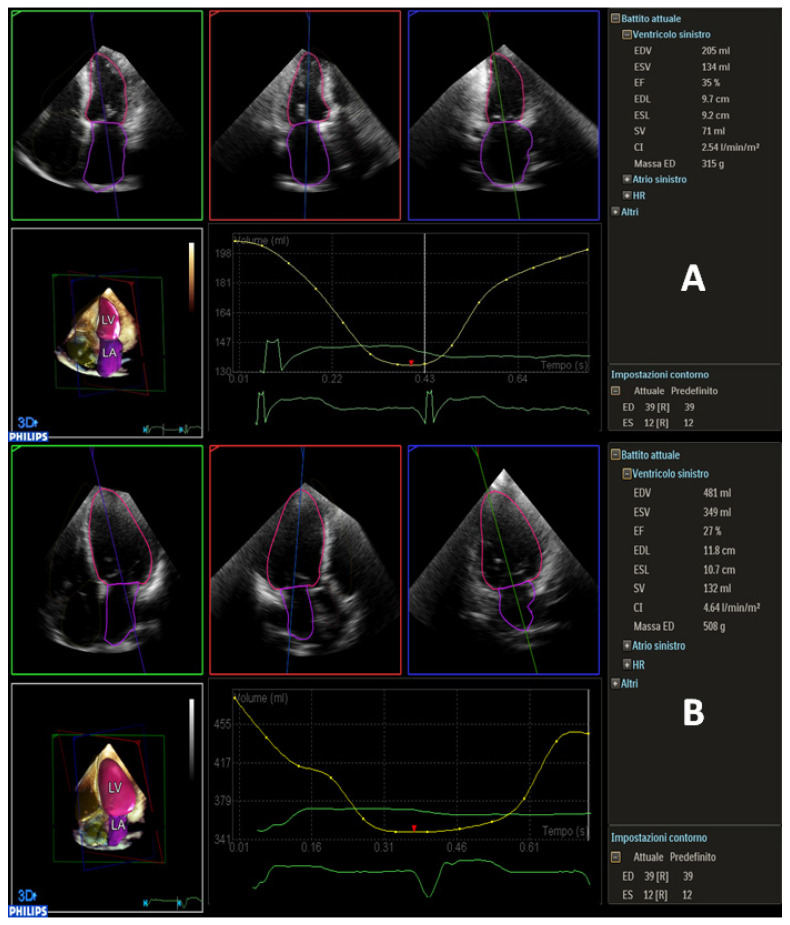
(**A**) Dynamic heart model analysis in D-MR patients, with LVEF of 35% and LVEDV of 205 mL. (**B**) Dynamic heart model analysis in P-MR patient with a severely dilated LV, LVEF of 27% and LVEDV of 480 mL.

**Figure 3 jcm-11-00645-f003:**
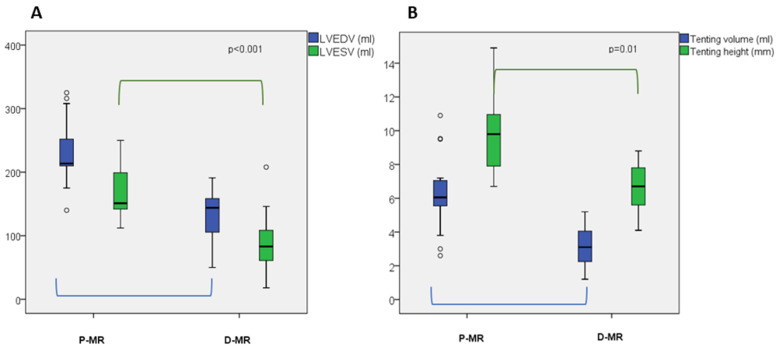
(**A**) Differences in mean LVEDV and LVESV between D-MR and P-MR patients, where *p* < 0.001 in both cases. (**B**) Differences in mean tenting height and tenting volume between D-MR and P-MR patients, where *p* = 0.01 in both cases. Box plots represent median, quartiles and extremes, and while circles represent the outliers.

**Figure 4 jcm-11-00645-f004:**
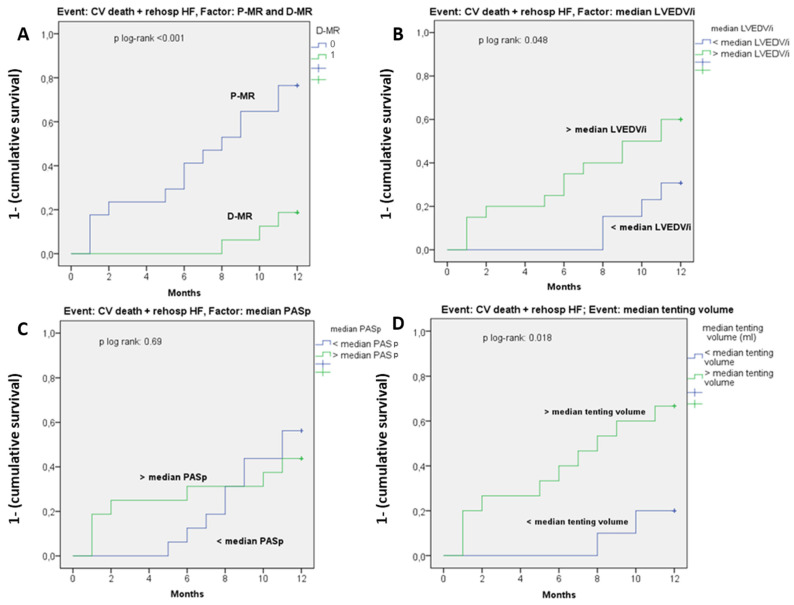
Survival analysis (Kaplan–Meier curves) for cardiovascular (CV) death and rehospitalization for HF. All curves show the 1-year event probability (expressed as 1—cumulative survival). (**A**) Factor: P-MR and D-MR status. (**B**) Factor: median value of LVEDV/I (mL/m^2^). (**C**) Factor: median value of PASp (mmHg) (**D**) Factor: median value of tenting volume (mL).

**Table 1 jcm-11-00645-t001:** Clinical and echocardiographic variables in the overall population and in Proportionate vs. Disproportionate MR patients at baseline.

Parameters	Total Cohort (*n* = 56)	Disproportionate MR (*n* = 28, 50%)	Proportionate MR (*n* = 28, 50%)	*p*
Clinical Features				
Age, (years)	73 ± 7	75 ± 7	70 ± 6	0.8
Euro Score	15.5 ± 10	15 ± 8	16 ± 13	0.7
STS Score	7.2 ± 8	9 ± 10	5 ± 5	0.3
Male sex, *n* (%)	41 (73)	20 (71)	21 (75)	0.7
Diabetes, *n* (%)	20 (37)	9 (32)	11 (39)	0.6
Hypertension, *n* (%)	48 (86)	25 (89)	23 (82)	0.9
Dyslipidemia, *n* (%)	36 (65)	16 (57)	20 (71)	0.3
Previous AMI, *n* (%)	28 (50)	12 (42)	16 (57)	0.2
Previous PCI, *n* (%)	22 (40)	9 (32)	13 (46)	0.2
Previous CABG, *n* (%)	11 (20)	4 (14)	7 (25)	0.3
CRF, *n* (%)	28 (50)	14 (50)	14 (50)	0.9
NYHA III-IV, *n* (%)	56 (100)	28 (100)	28 (100)	0.9
Nitrates, *n* (%)	25 (45)	11 (39)	14 (50)	0.08
ACE-inhibitors/ARBs, *n* (%)	33 (60)	16 (57)	17 (60)	0.8
Ivabradine, *n* (%)	2 (0.03)	0 (0)	2 (0.7)	0.2
Beta-blockers, *n* (%)	53 (95)	27 (96)	26 (92)	0.9
Anticoagulants, *n* (%)	12 (26)	4 (15)	8 (30)	0.07
Antiplatelets, *n* (%)	33 (58)	18 (69)	15 (57)	0.08
Aldosterone antagonists, *n* (%)	47 (85)	22 (78)	25 (89)	0.1
Diuretics, *n* (%)	56 (100)	28 (100)	28 (100)	0.9
Pacemaker, *n* (%)	14 (25)	5 (17)	9 (32)	0.08
AF, *n* (%)	11 (20)	4 (15)	7 (25)	0.07
Ischemic etiology, *n* (%)	28 (50)	12 (42)	16 (57)	0.2
Non-ischemic etiology, *n* (%)	28 (50)	16 (57)	12 (42)	0.2
Clips number (1), *n* (%)	29 (51)	15 (55)	14 (50)	0.8
Clips number (2) *n* (%)	26 (46)	13 (46)	13 (46)	0.9
Clips number (3) *n* (%)	1 (0.02)	0 (0)	1 (0.03)	0.9
Echocardiography				
EROA, cm^2^	0.3 (0.2–0.4)	0.3 (0.2–0.4)	0.31 (0.2–0.39)	0.7
Annulus AP diameter, mm	42 (29–50)	41 (29–50)	44 (38–50)	0.2
Annulus ellipticity, %	149 (112–170)	162 (155–170)	137 (112–160)	0.048
Tenting volume, mL	4.7 ± 2	3.2 ± 1.2	6.3 ± 2.3	0.01
Tenting height, mm	8 ± 1.9	6.5 ± 1.4	9.6 ± 2.4	0.01
PL angle, °	47 ± 12	48 ± 12	46 ± 10	0.6
AL angle, °	26 ± 9	24 ± 4	29 ± 8	0.08
LVEF, %	29 ± 8	32 ± 7	26 ± 5	0.003
LVEDV, mL	181 ± 49	135 ± 38	228 ± 48	<0.001
LVEDV/I, mL/m^2^	103 ± 28	80 ± 20	126 ± 27	<0.001
LVESV, mL	127 ± 42	88 ± 40	167 ± 41	<0.001
LVESV/I, mL/m^2^	77 ± 24	60 ± 20	94 ± 23	<0.001
LV mass, gr	275 ± 70	249 ± 63	301 ± 69	0.035
LA Vol/I, mL/m^2^	60 ± 26	58 ± 27	62 ± 25	0.065
RV ED area, cm^2^	18 ± 6	19 ± 5	18 ± 3	0.7
RV ES area, cm^2^	11 ± 4	11 ± 3	10 ± 5	0.6
RV FAC, %	47 ± 9	50 ± 8	45 ± 9	0.07
TAPSE, mm	21 ± 5	20 ± 3	21 ± 4	0.8
PASp, mmHg	42 ± 14	40 ± 16	45 ± 12	0.07
GLS, %	−8 (−11, −6)	−9 (−11, −7)	−7 (−8, −6)	0.048
Follow-up				
CV death/ HF rehospitalization, *n* (%)	15 (26)	2 (7)	13 (46)	<0.001
6-month echo, MR recurrence, *n* (%)	20 (35)	4 (14)	16 (57)	<0.001

ACE: angiotensin-converting enzyme, AF: atrial fibrillation, AL: anterior leaflet; AMI: acute myocardial infarction; AP: antero-posterior; ARBs: angiotensin II receptor blockers; CABG: coronary artery bypass grafting, CRF: chronic renal failure CV: cardiovascular, ED: end-diastolic, EROA: effective regurgitant orifice area, ES: end-systolic, FAC: fractional area change, GLS: global longitudinal strain, HF: heart failure, LA Vol/i: left atrium volume/BSA, LVEDV: left ventricular end-diastolic volume, LVEDV/i: left ventricular end-diastolic volume/BSA; LVEF: left ventricular ejection fraction, LVESV: left ventricular end-systolic volume; LVESV/i: left ventricular end-systolic volume/BSA, NYHA: New York Heart Association; PASp: pulmonary artery systolic pressure, PCI: percutaneous coronary intervention; RA: right atrium. RV: right ventricle, STS: Society of Thoracic Surgery Score, TAPSE; tricuspid annular plane excursion.

**Table 2 jcm-11-00645-t002:** Determinants of CV death/ HF rehospitalization within 1 year. Univariate analysis.

Variables	HR (IC 95%)	*p*
Age	0.96 (0.91–1)	0.19
Ischemic etiology	0.98 (0.4–2.3)	0.096
CRF	1.6 (0.66–1.18)	0.29
STS score	1.001 (0.94–1.1)	0.96
LVEDV/i, mL/m^2^	1.007 (1.001–1.015)	0.048
LVESV/i, mL/m^2^	1.005 (0.997–1.013)	0.23
LV mass g/m^2^	1.012 (0.997–1.028)	0.11
LVEF, %	1.002 (0.96–1.04)	0.94
PASp, mmHg	1.03 (1.01–1.055)	0.064
Annulus ellipticity, %	0.99 (0.98–1.01)	0.92
Annulus AP diameter, mm	0.96 (0.88–1.055)	0.45
Tenting volume (mL)	1.3 (1.08–1.57)	0.005
Tenting height (mm)	1.19 (0.995–1.4)	0.5
GLS, %	0.94 (0.7–1.2)	0.94
EROA, cm^2^	1.73 (0.39–7.4)	0.47
P-MR	3.4 (1.3–8.6)	0.009

Abbreviations as above.

**Table 3 jcm-11-00645-t003:** Intra and inter-observer variability of echocardiographic measurements.

Variable	Intra-Observer Agreement	Inter-Observer Agreement
LVEF, %	0.981 (0.92–0.996), *p* < 0.001	0.938 (0.75–0.986), *p* < 0.001
LVEDV/i, mL/m^2^	0.996 (0.985–0.999), *p* < 0.001	0.994 (0.736–0.999), *p* < 0.001
LVESV/i, mL/m^2^	0.998 (0.992–0.999), *p* < 0.001	0.997 (0.986–0.999), *p* < 0.001
MVQ analysis	0.998 (0.993–0.999), *p* < 0.001	0.996 (0.988–0.999), *p* < 0.001

Abbreviations as above, MVQ: mitral valve quantification.

## Data Availability

The data that support the findings of this study are available on reasonable request from the corresponding author.

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
