# Peer review of "3D Echo Characterization of Proportionate and Disproportionate Functional Mitral Regurgitation before and after Percutaneous Mitral Valve Repair"

_jcm, 2022, doi:10.3390/jcm11030645_

Round 1

Reviewer 1 Report

Dear Authors,

I was pleased to read the article entitled "3D Echo characterization of Proportionate and Disproportionate Functional Mitral Regurgitation before and after Percutaneous Mitral Valve Repair" The manuscript is written clearly and in an orderly manner. It discusses the addressed issues successively. However, I found a few issues that I propose to change.

  1. Minor issues
    1. Please provide full names of the authors, not initials
    2. Please correct typographical errors (no spaces, etc.)
  2. The main limitation of the study is a relatively small study group. According to this, please provide for non-normally distributed data rather median and min-max values than mean and SD
  3. Figure 2 – I suggest that the word "dinamic" should be replaced by the correct form: "dynamic"
  4. Figure 3. Please describe precisely what presents box and whisker plot: mean/median, SD/quartiles /extremes etc.?
  5. Figure 4. Please replace the term "1-cum survival" for more understandable for the reader. Please correct the description of lines on the first graph and unify PAS/PASp abbreviation on the third graph
  6. In Table 3 p-value for PASp is bolded, despite the nonsignificant level – please correct

It will be a pleasure to review the revised version of the article

Sincerely yours

Author Response

Please provide full names of the authors, not initials

A: we would like to thank the reviewer for the observation, authors’ full names are provided

Please correct typographical errors (no spaces, etc.)

A: we would like to thank the reviewer for the observation, typographical errors have been corrected

The main limitation of the study is a relatively small study group. According to this, please provide for non-normally distributed data rather median and min-max values than mean and SD

A: we added in Table 2 median and min-max (extremes) values as suggested for non-normally distributed variables as tested by Kolmogorov-Smirnov test.

Figure 2 – I suggest that the word "dinamic" should be replaced by the correct form: "dynamic"

A: we would like to thank the reviewer for the observation, we made the correction as suggested

Figure 3. Please describe precisely what presents box and whisker plot: mean/median, SD/quartiles /extremes etc.?

A: we would like to thank the reviewer for the observation, the box-plots represent median, quartiles and extremes; we added this information in the figure legend.

Figure 4. Please replace the term "1-cum survival" for more understandable for the reader. Please correct the description of lines on the first graph and unify PAS/PASp abbreviation on the third graph

A: we would like to thank the reviewer for the observation, we modified the figure and the figure legend as suggested.

In Table 3 p-value for PASp is bolded, despite the nonsignificant level – please correct

A: we would like to thank the reviewer for the observation. We bolded the value because there was a difference in the groups (although non-significant), however, we modified as suggested.

Reviewer 2 Report

Cimino et al have presented a manuscript entitled "3D Echo characterization of Proportionate and Disproportionate Functional Mitral Regurgitation before and after Percutaneous Mitral Valve Repair". In this study, the impact of PMVR is compared between patients with proportionate and disproportionate MR, investigating an extremely pertinent topic presently, following the monumental impact of the COAPT trial. They demonstrate a

The study is well written and presented very nicely. However, I have the following concerns:

  1. The number of patients is very low! That is the main caveat of this manuscript, as was well declared in the “limitations” section. Just as an example, the sentence “4) no other relevant clinical and echocardiographic features demonstrated to have prognostic implications over P-MR status” (line 231-233) is not accurate. You cannot make such claims in two groups of less than 30 patients. If there is any way to increase the number of patients- that would be very important.
  2. The cut-off of EROA/LVEDV = 0.14 needs more explanations. Can we do a sensitivity analysis? Discuss its AUC?
  3. I see no point in presenting a separate table for the full cohort. Either erase it or add a column to table 2
  4. There is no figure legend? After the title, figure 1 appears immediately.

Author Response

Reviewer 2.

  1. The number of patients is very low! That is the main caveat of this manuscript, as was well declared in the “limitations” section. Just as an example, the sentence “4) no other relevant clinical and echocardiographic features demonstrated to have prognostic implications over P-MR status” (line 231-233) is not accurate. You cannot make such claims in two groups of less than 30 patients. If there is any way to increase the number of patients- that would be very important.

A: we would like to thank the reviewer for the observation, we modified the text as suggested, giving less strength to out affirmation. Unfortunately, we have not significantly increased the number of enrolled patients in the last year because of CODIV 19-Outbreak, so we are not able to perform analysis on a wider sample.

  1. The cut-off of EROA/LVEDV = 0.14 needs more explanations. Can we do a sensitivity analysis? Discuss its AUC?

A: No further analysis was performed since this value has already been published and discussed (reference n. 11 and 29).

  1. I see no point in presenting a separate table for the full cohort. Either erase it or add a column to table 2

A: We would like to thank the reviewer for the observation. Despite we understand the suggestion, we preferred to not erase Table 1 since it shows additional minor information.

  1. There is no figure legend? After the title, figure 1 appears immediately.

A: we would like to thank the reviewer for the observation we made some changes and now you can find the figure legends immediately before their correspondent image.

Round 2

Reviewer 2 Report

Thank you for re-submitting. I understand the present constraints regarding patient number, and thank you for adding a figure legend. However, you did not really respond to comments 2 & 3 in a satisfactory manner. 

Author Response

Reviewer 2.

  1. The cut-off of EROA/LVEDV = 0.14 needs more explanations. Can we do a sensitivity analysis? Discuss its AUC?

We recognize that different methods have been proposed to classify MR in proportionate and disproportionate. Grayburn and colleagues performed a critical analysis on both COAPT and MITRA FR trials, introducing the conceptual framework of proportionate and disproportionate MR (P-MR and D-MR) (Grayburn et al. J Am Coll Cardiol Img 2019 12 (2) 353-362, currently reference 11). They proposed the cut-off of 0.14 for the EROA/LVEDV ratio to discriminate this two groups of patients (Packer M and Grayburn PA, JAMA Cardiol 2020; 5 (4):469-475). We now added this latter reference in the manuscript. We thank the reviewer for notice the missing reference.

Considering that this cut-off was obtained on a large and representative population, we decided to apply the same cut-off on our population. The limited number of patients enrolled in our study precludes the possibility to generate different cut-off.

  1. I see no point in presenting a separate table for the full cohort. Either erase it or add a column to table 2

We would like to thank the reviewer for the observation. We erased Table 1 adding a column to Table 2, which become table 1. We modified the text and caption accordingly.
